# Effect of Thermomechanical Treatments on Microstructure, Phase Composition, Vickers Microhardness, and Young's Modulus of Ti-xNb-5Mo Alloys for Biomedical Applications

**Giovana Collombaro Cardoso** [1], **Marília Afonso Rabelo Buzalaf** [2], **Diego Rafael Nespeque Correa** [1,3] **and Carlos Roberto Grandini** [1,*]

1 Laboratório de Anelasticidade e Biomateriais, UNESP—Universidade Estadual Paulista, Bauru 17033-360, SP, Brazil; giovana.collombaro@unesp.br (G.C.C.); diego.correa@ifsp.edu.br (D.R.N.C.)
2 Bauru School of Dentistry, University of São Paulo, Bauru 17012-901, SP, Brazil; mbuzalaf@fob.usp.br
3 Science, and Technology of São Paulo, IFSP—Federal Institute of Education, Sorocaba 18095-410, SP, Brazil
* Correspondence: carlos.r.grandini@unesp.br

**Abstract:** The development of new β-Ti alloys has been extensively studied in the medical field in recent times due to their more suitable mechanical properties, such as a relatively low Young's modulus. This paper analyzes the influence of heat treatments (homogenization and annealing) and hot rolling on the microstructure, phase composition, and some mechanical properties of ternary alloys of the Ti-xNb-5Mo system, with an amount of Nb varying between 0 and 30 wt%. The samples are produced by argon arc melting. After melting, the samples are homogenized at 1000 °C for 24 h and are hot rolled and annealed at 1000 °C for 6 h with slow cooling. Structural and microstructural analyses are made using X-ray diffraction and optical and scanning electron microscopy. Mechanical properties are evaluated by Vickers microhardness and Young's modulus. The amount of β phase increases after heat treatment and reduces after hot rolling. The microhardness and Young's modulus of all heat-treated samples decrease when compared with the hot rolled ones. Some samples exhibit atypical Young's modulus and microhardness values, such as 515 HV for the as-cast Ti-10Nb-5Mo sample, indicating the possible presence of ω phase in the microstructure. The Ti-30Nb-5Mo sample suffers less variation in its phase composition with thermomechanical treatments due to the β-stabilizing effect of the alloying elements. The studied mechanical properties indicate that the annealed Ti-30Nb-5Mo sample has potential for biomedical applications, exhibiting a Young's modulus value of 69 GPa and a microhardness of 236 HV.

**Keywords:** biomaterial; Ti alloys; thermomechanical treatments; phase composition; microstructure; microhardness; Young's modulus

## 1. Introduction

Currently, the use of biomaterials is growing and plays a crucial role in the biomedical area. In recent years, materials processing technologies have advanced considerably, allowing for the development of numerous biomaterials with adaptable properties for several applications, such as orthopedic, dental, and cardiovascular implants; drug delivery; and tissue engineering [1–3].

Commercially pure titanium (CP-Ti) and the Ti-6Al-4V alloy are widely used as metallic biomaterials for orthopedic and dental bone implants [4,5]. However, these materials have a higher Young's modulus than those of the cortical human bones (<30 GPa) [4,6] and potentially toxic elements (V and Al) [6–8]. Therefore, efforts have been conducted to develop new Ti alloys without cytotoxic elements, containing, for example, Mo, Nb, Ta, and Zr, and with mechanical properties closer to those of the human bones [6]. Each Ti phase has different mechanical properties, with the β phase having lower elastic modulus values. For this reason, new Ti alloys are developed using β-stabilizing elements [9–11].

Ideally, a biomaterial for an implant must have a non-toxic composition, light weight, adequate mechanical strength, good castability, low Young's modulus, and high corrosion and wear resistance. In Ti's solid solutions, it is possible to improve these properties by modifying the type and concentration of the alloying elements [12,13]. When added to Ti alloys, Nb can increase corrosion resistance [14], whereas Mo can enhance mechanical compatibility, both being recognized as β-stabilizer elements [11,14,15]. Furthermore, they are known non-toxic or allergenic metals that can improve the properties of Ti alloys [11–13,16].

In addition to chemical composition, thermomechanical processing can also change the structure, microstructure, and mechanical properties of Ti and its alloys [9,17,18]. Heat treatments can provide stress relief, minimize imperfections, and promote a more homogeneous phase distribution [9,19]. Moreover, thermomechanical treatments, such as hot rolling, can increase mechanical and wear resistance, reduce ductility, and induce metastable α″ phase formation [18,20,21].

In this study, the influence of some thermomechanical processes on the crystalline structure, microstructure, and some mechanical properties of Ti-xNb-5Mo (x = 0, 10, 20, 30 wt%) aiming toward biomedical applications were investigated. The phase composition, microstructure, and mechanical properties of the samples were evaluated after argon arc melting, hot rolling, and homogenization and annealing heat treatments. Although some articles point out that Mo can be toxic [8,22], there are works showing that Ti-alloys containing Mo exhibit excellent biocompatibility [23], such as Ti-15Mo [24], Ti-15Mo-5Mn [25], and Ti-12Mo-6Zr-2Fe (TMZF) [26]. As it is a strong β-stabilizing element, high Mo contents also tend to increase bond strength between atoms, which may increase the Young modulus [18]. Thus, to avoid possible cytotoxicity of Mo and to not negatively influence the mechanical properties, it was chosen to work with a low concentration of this element. The Nb concentrations were chosen to study alloys with α + β phases, up to the alloy with only a β phase.

Additionally, Brazil is one of the leading countries in producing and concentrating Nb minerals, accounting for about 90% of all world production, and Companhia Brasileira de Metalurgia e Mineração (CBMM) is the world's leading producer of this element [27,28]. Moreover, niobium is of interest in the biomedical area because it increases corrosion resistance, reduces Young's modulus, and promotes shape memory behavior in Ti alloys, aside from Ti alloys being biocompatible, non-allergic, and non-toxic [29,30].

## 2. Materials and Methods

Ingots of Ti-xNb-5Mo (x = 0, 10, 20, 30 wt%) alloys, weighing approximately 60 g each, were produced by arc melting in an argon gas atmosphere, using commercially pure Ti grade 2 bars (CP-Ti, Sandinox, Sorocaba, Brazil), Mo wires (99.9% purity, Sigma-Aldrich, Saint Louis, MO, USA), and Nb chips (99.8% purity, Sigma-Aldrich) as precursor materials. The details of casting, chemical composition, structure and microstructure, and selected mechanical properties of as-cast alloys are described in a previous paper [31]. After melting, the ingots were homogenized by a heat treatment at 1273 K in a vacuum of $10^{-7}$ Torr for 24 h, with slow cooling to reduce internal stresses and imperfections and to eliminate possible agglomerated or segregated elements. After that, the samples were hot rolled at 1273 K, without a controlled atmosphere, to obtain samples in a regular format for further analysis. A reduction of approximately 1 mm in thickness per pass, until reaching a final thickness of approximately 4 mm, was used. Finally, an annealing heat treatment was conducted at 1273 K in a vacuum of $10^{-7}$ Torr for 6 h to reduce the internal stresses of the material and to eliminate possible dislocations caused by mechanical deformation.

X-ray diffractometry (XRD) was used to evaluate structure and phase composition. The measurements were made using Panalytical X'Pert-Pro model equipment (Malvern Panalytical, Malvern, UK), with Cu-Kα radiation, operated at 30 mA and 40 kV, and with continuous-time mode. Rietveld refinement of the XRD diffractograms was performed using GSAS software [32] with the EXPEGUI interface [33]. The crystallographic datasheets of α and β Ti-phases [34] were used. To eliminate the experimental contributions of

the equipment, a standard CP-Ti sample was used [18]. Microstructural information was obtained by optical microscopy (OM), using an Olympus BX51M model equipment (Olympus, Tokyo, Japan), and scanning electron microscopy (SEM) by a Carl Zeiss EVO-015 model microscope (Carl Zeiss, Oberkochen, Germany) in the secondary electron (SE) imaging mode.

Microhardness measurements were obtained using Shimadzu HMV-2 model equipment. Five indentations made performed in each sample, using a load of 25 gf (0.245 N) for 10 s, based on the ASTM E92 standard [35]. Following the standard procedures, Young's modulus values were obtained by the impulse excitation method, with Sonelastic® equipment (ATCP Physical Engineering, São Carlos, Brazil) [36]. Average values were calculated from three distinct specimens for each sample, with five measurements on each specimen.

## 3. Results and Discussion

The diffractograms of the produced as-cast, homogenized, hot rolled, and annealed samples are shown in Figure 1. The phase compositions quantified by the Rietveld refinement of each sample are shown in Figure 2. The as-cast Ti-5Mo sample showed coexistence of the $\alpha'$ (63%), $\alpha''$ (22%), and $\beta$ (15%) phases. After the homogenization heat treatment, there was an increase in the $\alpha'$ phase (82%) and a reduction in the amount of $\alpha''$ phase (1%), and the $\beta$ phase remained almost unchanged (17%). The hot rolling process decreased the amount of $\beta$ phase to 5%, whereas $\alpha''$ phase was raised close to its original amount (22%). The annealing treatment augmented the quantity of $\beta$ phase (12%) and diminished the amount of the martensitic $\alpha'$ and $\alpha''$ phases (73% and 15%, respectively). For the Ti-10Nb-5Mo sample, the $\alpha'$ and $\beta$ phase proportions elevated from 12% and 38% to 44% and 50%, respectively, whereas the $\alpha''$ phase decayed from 50% to 6%, between the as-cast and homogenized conditions. After hot rolling, the amount of $\alpha'$ phase had a slight decrease (38%) along with the $\beta$ phase (25%), whereas the $\alpha''$ phase reached 37%. After annealing, the phase composition returned to similar amounts compared to the as-cast conditions for the Ti-20Nb-5Mo alloy, whose phase composition after melting consisted of 40% $\alpha''$ phase and 60% $\beta$ phase. The sample exhibited an increase in the $\beta$ phase (69%) after homogenization heat treatment, whereas in the hot rolled condition this value decreased to 49%. The Ti-30Nb-5Mo samples showed no changes in phase composition after thermomechanical processing, presenting only the $\beta$ phase in as-cast, homogenized, and annealed conditions. After hot rolling, a minor peak of $\alpha''$ phase was identified, induced by mechanical deformation. However, the amount of this phase was not enough to be measured properly.

Due to overlapping phases, some XDR lines were wider than usual, especially after the melting and hot rolling of the Ti-10Nb-5Mo alloy samples. Therefore, the Rietveld method was used to analyze the diffractogram to separate, identify, and quantify each phase.

Overall, the heat treatments facilitated the growth of $\alpha'$ and $\beta$ phases and suppressed $\alpha''$ phase precipitation due to the recrystallization of the microstructure during slow cooling and stress relieving. On the other hand, hot rolling diminished the $\alpha'$ and $\beta$ phases and favored $\alpha''$ phase precipitation, due to high cooling rates in addition to the defects promoted by mechanical deformations [18].

The microstructural features of the Ti-5Mo, Ti-10Nb-5Mo, Ti-20Nb-5Mo, and Ti-30Nb-5Mo samples for each thermomechanical condition are shown in Figures 3–6, which corroborate directly with the XRD results. The images of the Ti-5Mo and Ti-10Nb-5Mo samples depict similar lamellar structures. Coarser needles, which are typical of $\alpha'$ phase, as well as some acicular structures of $\alpha''$ phase and grain boundary characteristics of $\beta$ phase appeared in both samples [37]. The Ti-20Nb-5Mo sample displayed noticeable $\beta$ phase grain boundaries in all conditions and minor amount of $\alpha''$ phase needles inside the grains of the homogenized and annealed conditions. Lastly, the Ti-30Nb-5Mo sample exhibited only $\beta$ phase grain boundaries in all conditions. The micrographs of the hot rolled samples present a clear deformation at the grain in the rolling direction, remaining smaller and elongated, whereas the heat treatments increased the size of the grains due to recrystallization

and stress relief mechanisms at high temperatures [18,38]. More significant changes in the microstructures of the samples of Ti-5Mo and Ti-10Nb-5Mo alloys were observed due to a greater amount of α phase.

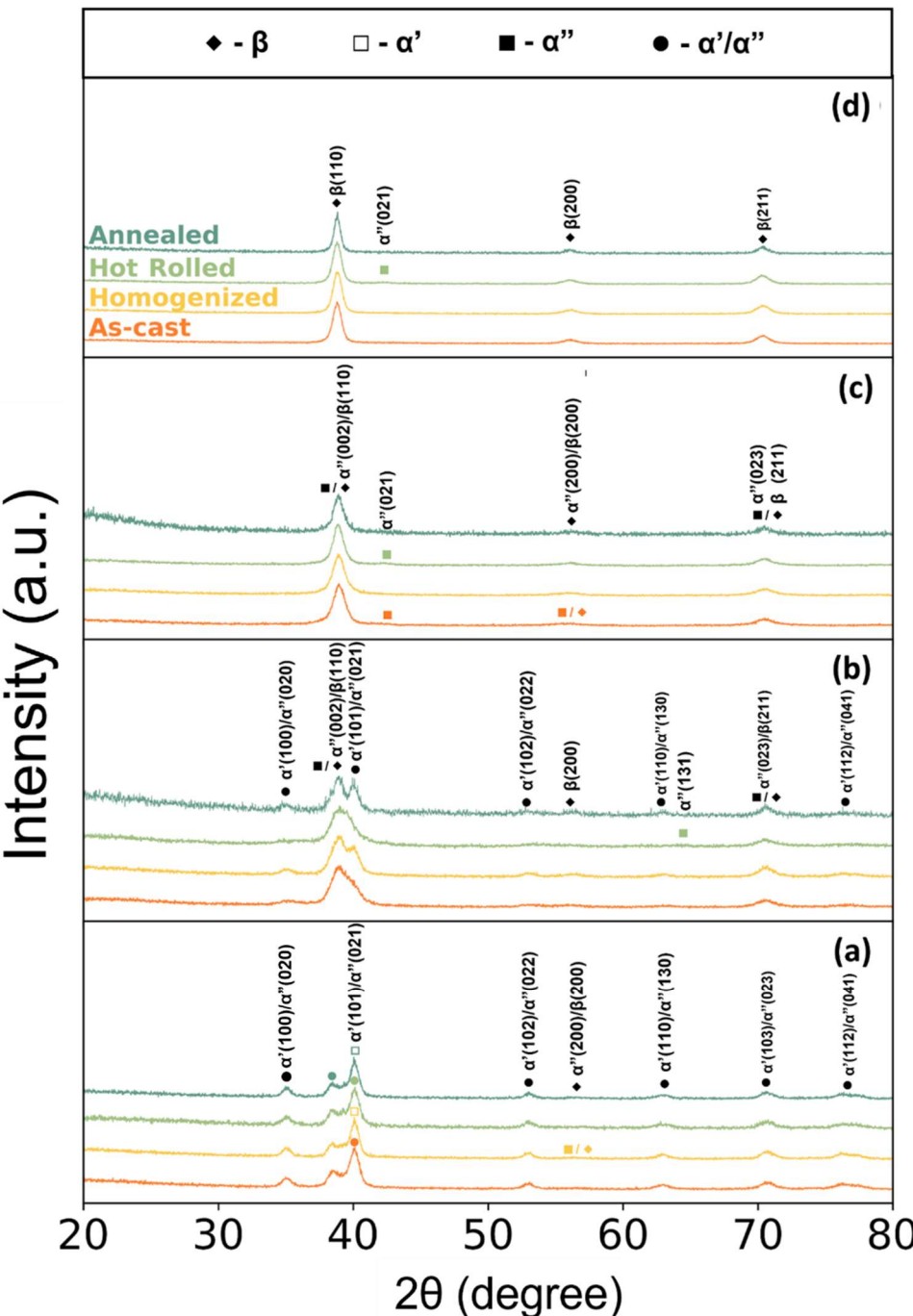

**Figure 1.** XRD diffractograms of the Ti-5Mo (**a**), Ti-10Nb-5Mo (**b**), Ti-20Nb-5Mo (**c**), and Ti-30Nb-5Mo (**d**) samples after thermomechanical processing.

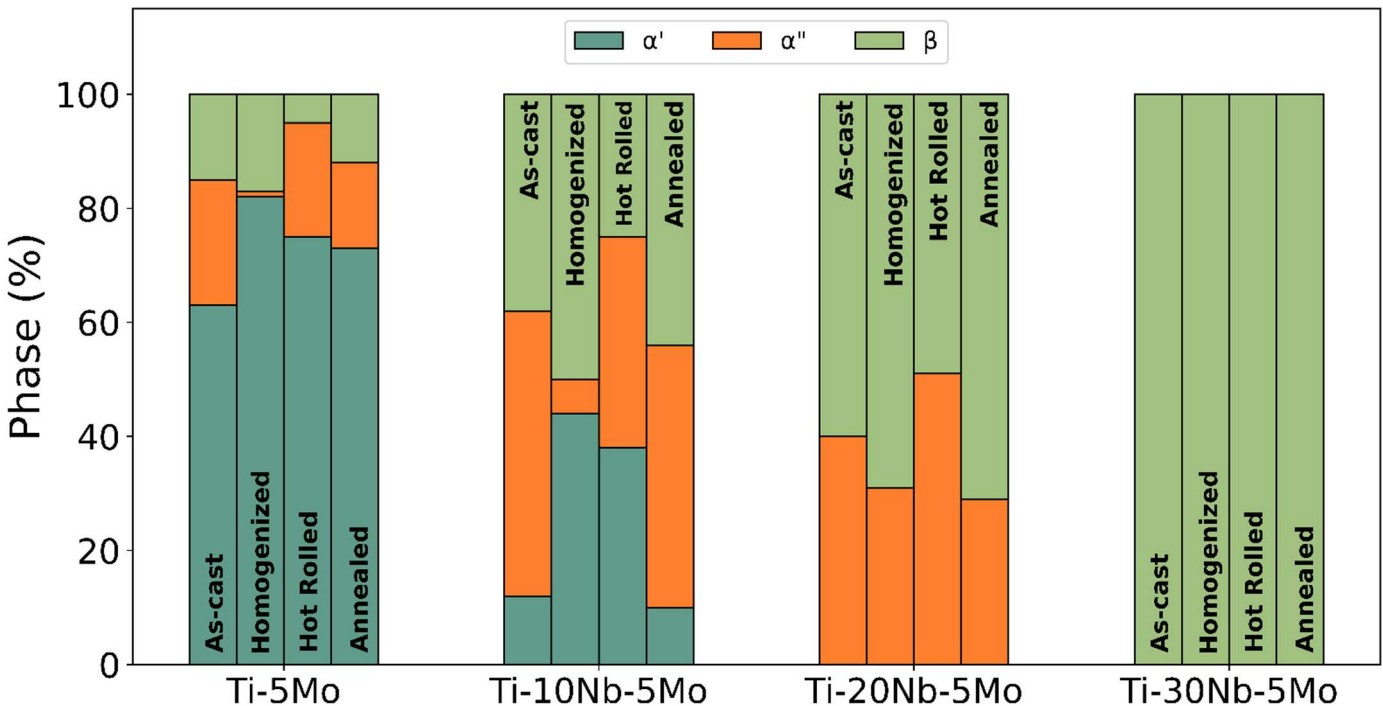

**Figure 2.** Phase composition of the Ti-xNb-5Mo samples, obtained by Rietveld's refinement.

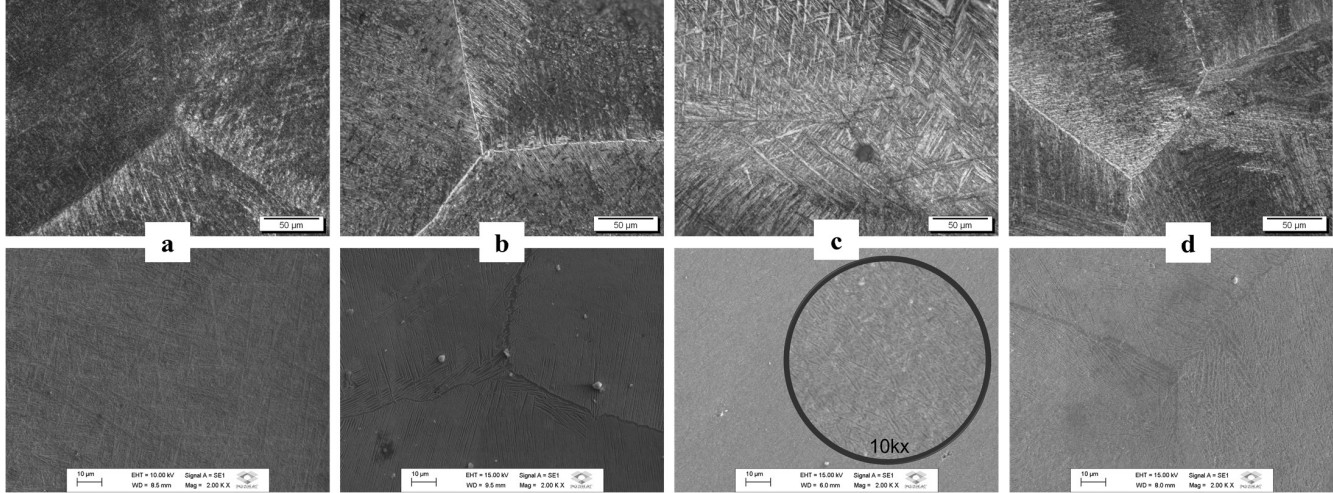

**Figure 3.** Optical micrographs (**above**) and SEM micrographs (**below**) in as-cast (**a**), homogenized (**b**), hot rolled (**c**), and annealed (**d**) conditions, for Ti-5Mo samples.

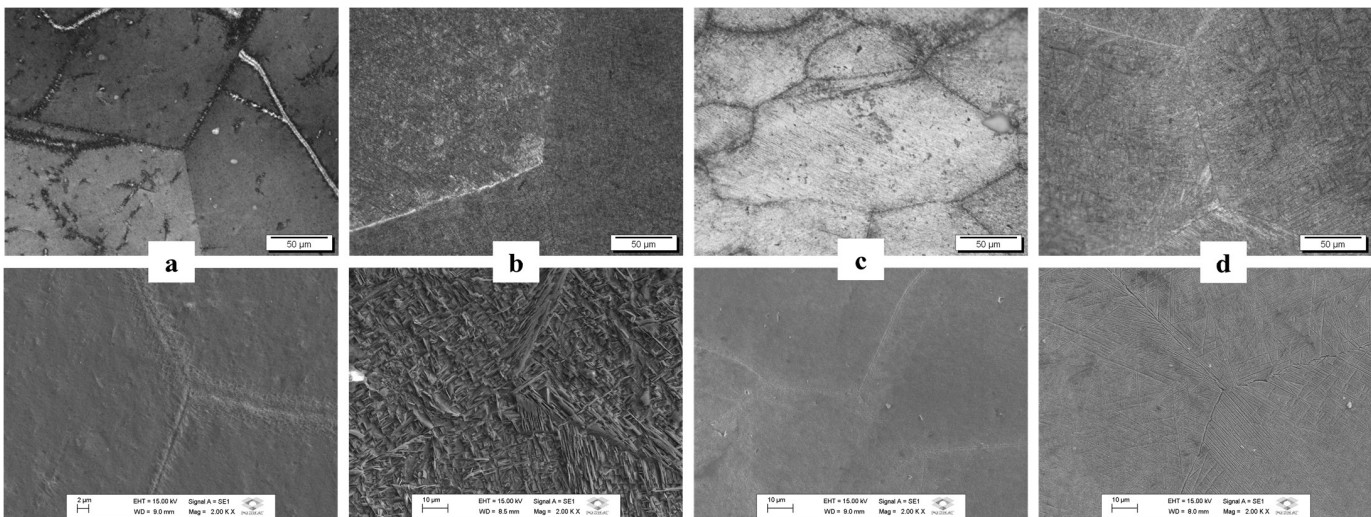

**Figure 4.** Optical micrographs (above) and SEM micrographs (below) in as-cast (**a**), homogenized (**b**), hot rolled (**c**), and annealed (**d**) conditions, for Ti-10Nb-5Mo samples.

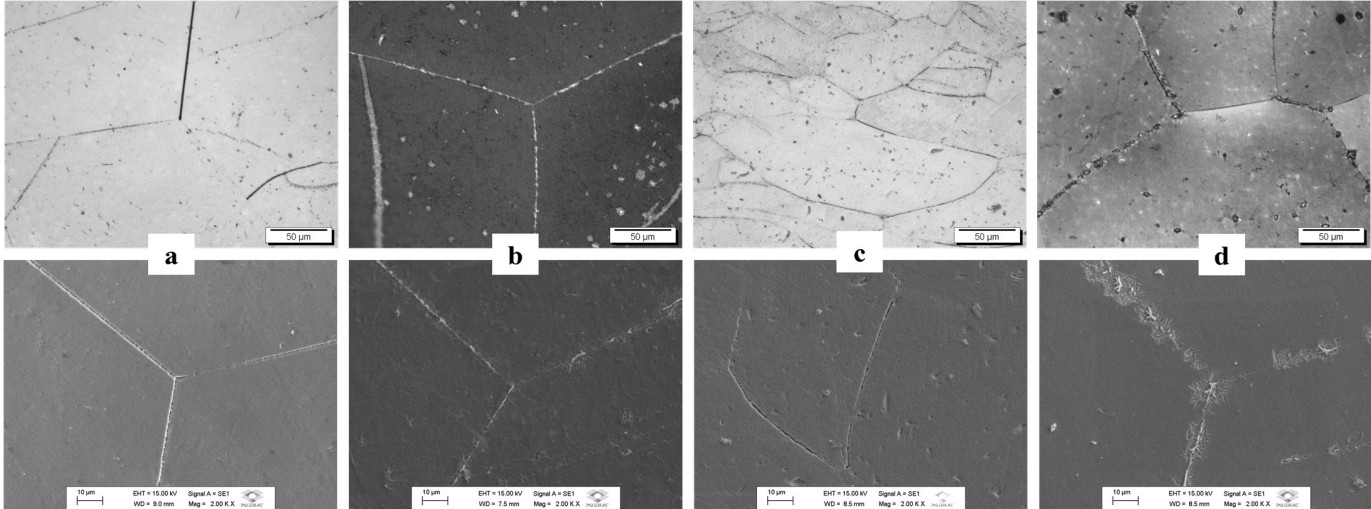

**Figure 5.** Optical micrographs (above) and SEM micrographs (below) in as-cast (**a**), homogenized (**b**), hot rolled (**c**), and annealed (**d**) conditions, for Ti-20Nb-5Mo samples.

The Vickers microhardness values of the studied samples are displayed in Figure 7. Due to a solid solution and the phase precipitation hardening effect, all samples showed higher values than CP-Ti grade 2 (148 HV) [39]. As-cast and hot rolled Ti-5Mo and Ti-20Nb-5Mo, homogenized and annealed Ti-10Nb-5Mo, and hot rolled Ti-30Nb-5Mo samples showed similar values to those of Ti-6Al-4V (304 HV) and AISI 316L (289 HV). In general, hot rolling tended to raise the microhardness values of the samples as a result of increases in dislocation density by mechanical working [40], whereas heat treatments caused decay due to the stress relief. However, this trend was not observed for the Ti-20Nb-5Mo sample, which had an elevated microhardness after heat treatment, possibly due to the formation of the metastable ω phase, which makes the material harder and more brittle [16,41,42]. Thoemmes et al. [43] found similar results for heat treatment at 1000 °C for 24 h slowly cooled Ti-29Nb samples and concluded that the cooling rate may have been slow enough that the samples were subjected to an aging heat treatment. The ω phase may also be present in the as-cast Ti-10Nb-5Mo sample, which may explain the significant increase in the values of this sample. As-cast, homogenized, and annealed Ti-30Nb-5Mo samples presented better hardness values to be used as biomaterials.

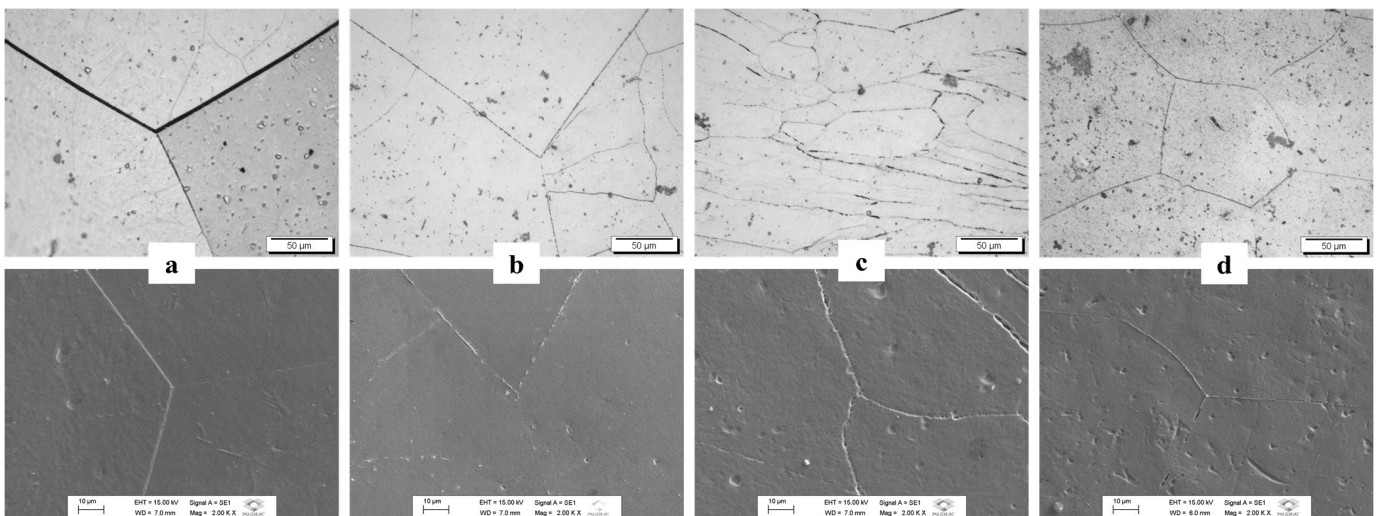

**Figure 6.** Optical micrographs (above) and SEM micrographs (below) in as-cast (**a**), homogenized (**b**), hot rolled (**c**), and annealed (**d**) conditions, for Ti-30Nb-5Mo samples.

Figure 8 compares the Young's modulus values of hot rolled and annealed samples. The Young's modulus values were influenced by the annealing treatment, except for the Ti-5Mo sample, whose value remained at around 100 GPa in both conditions. The Ti-10Nb-5Mo-10Nb and Ti-30Nb-5Mo samples had their values decreased from 98 GPa and 73 GPa to 78 GPa and 69 GPa, respectively, due to the increase in β phase quantity between the hot rolled and annealed samples. On the other hand, the values of the annealed Ti-20Nb-5Mo sample increased in comparison to those of the hot rolled condition, which is another indication of ω phase presence in this sample. In general, the ω phase in Ti alloys has the highest modulus values, followed by α, α′, α″, and β phases [9,10,20,44].

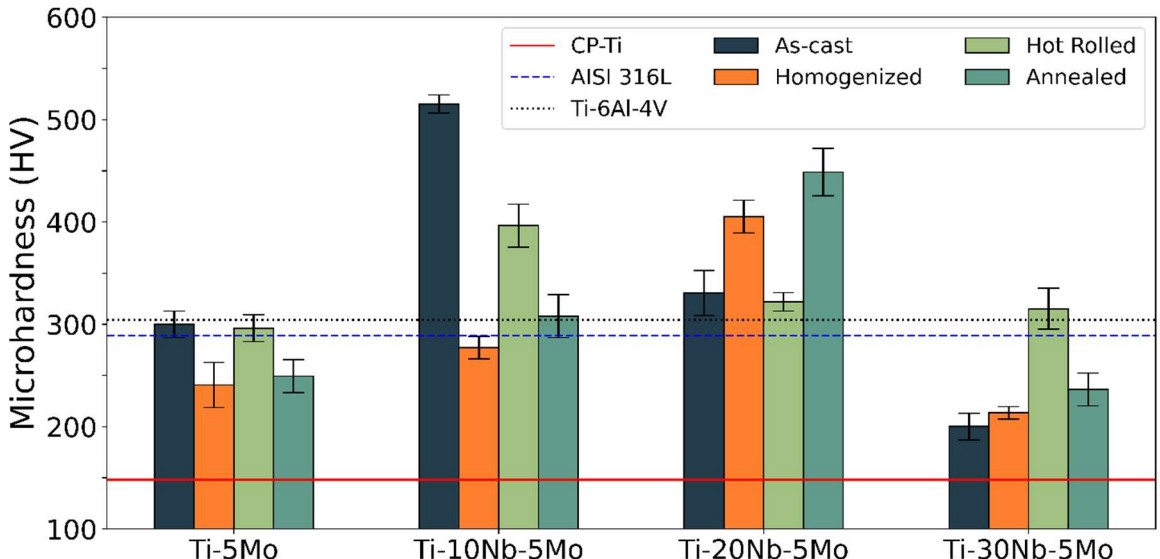

**Figure 7.** Vickers microhardness of the Ti-xNb-5Mo samples as a function of thermomechanical processing.

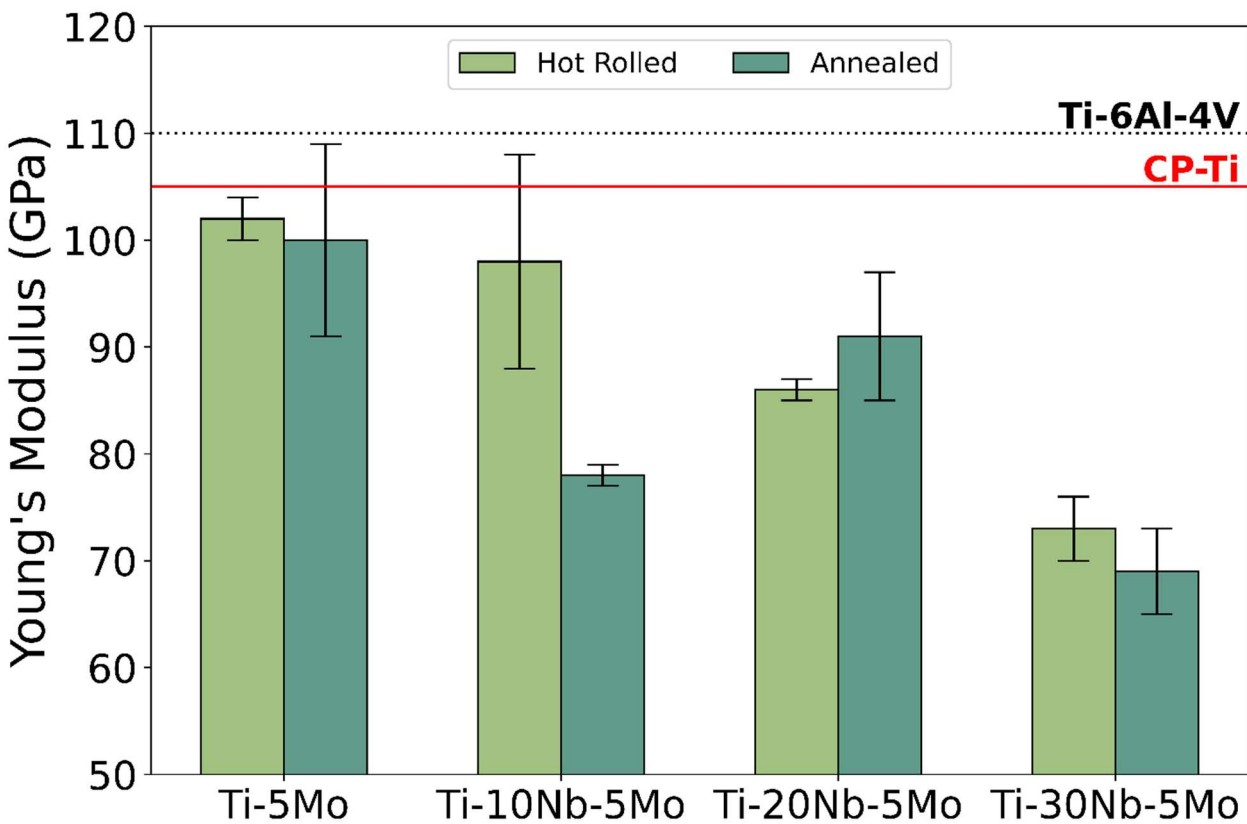

**Figure 8.** Young's Modulus of the Ti-xNb-5Mo samples as a function of thermomechanical processing.

Different thermomechanical treatments and levels at which plastic deformation is applied both affected deformation mechanisms, causing a change in Young's modulus [45–47]. The Ti-30Nb-5Mo samples suffered few changes in their microstructure, leading to a minor variation in their modulus.

The different compositions of the samples also influenced the variation of the mechanical properties studied as a function of the thermomechanical treatments. The sample with 10% Nb was more susceptible to variations in its Young's Modulus and microhardness with different processing.

The produced Ti-xNb-5Mo alloys have better Young's modulus values than some commercial alloys, such as CP-Ti grade 2 and Ti-6Al-4V. The annealed Ti-30Nb-5Mo sample exhibits the lowest Young's modulus (69 GPa) among all produced samples, with a value around two times higher than that of human cortical bone [48]. The microhardness of this same sample is lower than those of other metallic biomaterials, such as Ti-6Al-4V and AISI 316L, which is another appropriate property for biomedical applications, since it can be easily mechanically formed.

### 4. Conclusions

With the results presented above, it is possible to conclude that:

- The microstructures of the samples were sensitive to the thermomechanical processing performed. In the Ti-5Mo, Ti-10Nb-5Mo, and Ti-20Nb-5Mo samples, the homogenization heat treatment promoted the growth of $\alpha'$ and $\beta$ phases and the reduction in $\alpha''$ phase, whereas hot rolling suppressed $\alpha'$ and $\beta$ phases and facilitated $\alpha''$ phase formation. With a majority of $\beta$ phase, the Ti-30Nb-5Mo sample showed a minor amount of $\alpha''$ phase, not quantified by the Rietveld method, after hot rolling.
- The studied mechanical properties of the alloy samples with 10% Nb were more sensitive to the thermomechanical treatments.

- The microhardness of all samples was higher than that of CP-Ti, leading to increased wear resistance.
- The high Vickers microhardness and Young's modulus of the as-cast and heat-treated Ti-10Nb-5Mo and Ti-20Nb-5Mo samples may indicate the presence of ω phase in their microstructures.
- All samples exhibited a lower Young's modulus than those of commercial alloys.
- Annealed Ti-30Nb-5Mo samples showed the best values of Young's modulus (69 GPa), presenting a favorable mechanical property for biomedical applications.

**Author Contributions:** Conceptualization, G.C.C.; methodology, G.C.C., M.A.R.B. and D.R.N.C.; investigation, G.C.C. and D.R.N.C.; resources, M.A.R.B., D.R.N.C. and C.R.G.; data curation, M.A.R.B., D.R.N.C. and C.R.G.; writing—original draft preparation, G.C.C.; writing—review and editing, M.A.R.B., D.R.N.C. and C.R.G.; supervision, C.R.G.; project administration, C.R.G.; funding acquisition, C.R.G. All authors have read and agreed to the published version of the manuscript.

**Funding:** This research was funded by Coordenação de Aperfeiçoamento de Pessoal de Nível Superior—Brazil (CAPES)—Finance Code 001; CNPq (grant # 308.204/2017-4) and FAPESP (grant #2015/50.280-5).

**Institutional Review Board Statement:** Not applicable.

**Informed Consent Statement:** Not applicable.

**Data Availability Statement:** The data that support the findings of this study are available from the corresponding author.

**Acknowledgments:** The authors would like to thank Oscar Balancin and Rover Belo (UFSCar) for using hot rolling equipment.

**Conflicts of Interest:** The authors declare that they are not aware of competition for financial interest or personal relationships that may have influenced the work reported in this paper.

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
