# Peer review of "Effect of Thermomechanical Treatments on Microstructure, Phase Composition, Vickers Microhardness, and Young’s Modulus of Ti-xNb-5Mo Alloys for Biomedical Applications"

_metals, doi:10.3390/met12050788_

Round 1

Reviewer 1 Report

Please do the following corrections.

Line 24: Typo error "atypical"

Line 40: Spelling error "als"

Section: Materials and Methods

Lines 71-78: Try to summarize information about heat treatment and annealing process conditions in Table.

It is very hard to follow in text.

As you did heat treatment and annealing process, it will be good to add results about the chemical compositions of tested samples by EDS(X) method. It will help readers to understand the effect of chemical compositions in the treated samples.  

Figure 3/4/5/6: Highlight the microstructure features in the image. For ex. lamellar structures, coarser needles, acicular structures, etc.

Add more information about Young's modulus calculation from tensile test experiments.

Looking forward to the modified manuscript.

The paper can be accepted after the modifications.

Thank You.

Author Response

We would like to thank the reviewers and Academic Editor for their careful reading and suggestions. We seek to meet all of them, which will enrich the paper. The pertinent changes can be found in the revision mode in the text.

Below we are answering all presented questions.

Reviewer 1:

Please do the following corrections.

Thank you very much for the revision. Indeed, it will improve the manuscript.

Line 24: Typo error "atypical"

We do not understand. Atypical is an adjective not representative of a type, group, or class. The correct sense presented is “a not typical Young’s modulus and Hardness”.

Line 40: Spelling error "als"

This occurred because of an automatic word separation by the template of MDPI. The correct word is “materials”, and the template separates the word “materi-als”.

Lines 71-78: Try to summarize information about heat treatment and annealing process conditions in Table. It is very hard to follow in text.

We understand that putting the information in a Table is more complicated. We prefer in the present form.

As you did heat treatment and annealing process, it will be good to add results about the chemical compositions of tested samples by EDS(X) method. It will help readers to understand the effect of chemical compositions in the treated samples.

We measured EDS only after the melting. We can measure the EDS in all conditions, but it is impossible in the time established by the editor.

Figure 3/4/5/6: Highlight the microstructure features in the image. For ex. lamellar structures, coarser needles, acicular structures, etc.

The Figs. were modified.

Add more information about Young's modulus calculation from tensile test experiments.

The measurements of Young's modulus follow ASTM E1876-09, Standard Test Method for Dynamic Young’s Modulus, Shear Modulus, and Poisson’s Ratio by Impulse Excitation of Vibration. Following the standard procedures, Young's modulus values were obtained using the impulse excitation method with Sonelastic® equipment (ATCP Physical Engineering).

Sincerely yours,

Prof. Carlos Roberto Grandini, FBSE

Reviewer 2 Report

In the work, the influence of heat treatments and hot rolling on the microstructure, phase composition, and some mechanical properties of ternary alloys of the Ti-5Mo-Nb system with the amount of Nb varying was studied aiming to biomedical applications. The reviewer believes that this is a paper of high quality. It is suggested that the manuscript be accepted after the following comments are taken care of metals.

  1. Sound and concise conclusions and abstract with the supporting results should be given to the reader. The academic language should be refined by proof-read carefully. Some ambiguous points should be deleted.
  2. It just supplies some experimental results but no deeply analysis. The in-depth discussion should be added.
  3. The combination of cooling rate, temperature, mechanical deformation, and recrystallization processes of the thermomechanical treatments led to significant variations in Vickers microhardness and Young's modulus values of the samples. For some alloys, the severe plastic deformation contributed to solid-state and microstructural modification has inhibited the growth of recrystallized grains effectively with fine-grain microstructures to realize high strengthening-toughening efficiency [https://doi.org/10.1016/j.coco.2021.100776]. The deformation-driven metallurgy with high plastic strain and frictional/deformation heat contributes to dynamic recovery and recrystallization [https://doi.org/10.1016/j.compscitech.2021.109225]. The direct supporting results with detail discussion should be added.

Author Response

We would like to thank the reviewers and Academic Editor for their careful reading and suggestions. We seek to meet all of them, which will enrich the paper. The pertinent changes can be found in the revision mode in the text.

Below we are answering all presented questions.

In the work, the influence of heat treatments and hot rolling on the microstructure, phase composition, and some mechanical properties of ternary alloys of the Ti-5Mo-Nb system with the amount of Nb varying was studied aiming to biomedical applications. The reviewer believes that this is a paper of high quality. It is suggested that the manuscript be accepted after the following comments are taken care of metals.

Thank you very much for the revision. Indeed, it will improve the manuscript.

Sound and concise conclusions and abstract with the supporting results should be given to the reader.

We improved the abstract and the conclusions.

The academic language should be refined by proof-read carefully. Some ambiguous points should be deleted.

We made a complete revision of the paper, improving the language.

It just supplies some experimental results but no deeply analysis. The in-depth discussion should be added.

We improved the discussion based on the obtained results and the literature.

The combination of cooling rate, temperature, mechanical deformation, and recrystallization processes of the thermomechanical treatments led to significant variations in Vickers microhardness and Young's modulus values of the samples. For some alloys, the severe plastic deformation contributed to solid-state and microstructural modification has inhibited the growth of recrystallized grains effectively with fine-grain microstructures to realize high strengthening-toughening efficiency [https://doi.org/10.1016/j.coco.2021.100776]. The deformation-driven metallurgy with high plastic strain and frictional/deformation heat contributes to dynamic recovery and recrystallization [https://doi.org/10.1016/j.compscitech.2021.109225]. The direct supporting results with detail discussion should be added.

We introduced these two suggested references and used them in the improvement of the discussion of the results,

Sincerely yours,

Prof. Carlos Roberto Grandini, FBSE

Reviewer 3 Report

The manuscript is “some” research on the “effect of some thermomechanical treatments” of Ti-5Mo-Nb. The figures are well-designed, however all level of research and manuscript is very low. It is not clear why do authors use exactly these compositions, these experiments performed and why these conclusions are important for the readers. The authors are recommended to repeat the experiment with good quality ingots and rewrite the article and resubmit it next time. Before the resubmission, please clarify the following drawbacks:

  • The scientific language should be used for the manuscript. For example the title now has word “some” instead of “various” and list of all activities performed by the authors, without any scientific significance and reason: “Effect of some thermomechanical treatments on microstructure, phase composition, Vickers microhardness, and Young's modulus of Ti-5Mo-Nb alloys for biomedical applications“.
  • Since the alloys contain up to 30% of Nb, they should be named Ti-xNb-5Mo. It is common to mention the alloying elements with higher amount first.
  • It is not described in introduction why do authors use alloys with exactly from 0 to 30% of Nb.
  • Impurities and compositions of ingots are not studied.
  • Low-quality alloying elements are used (for example CP-Ti). It makes sense for large ingots, but not for 60g laboratory ingots. New pure ingots should be melted and studied. The impurities level and uniformity of composition should be evaluated.
  • Why do XRD lines are so wide in Fig.1 especially after annealing and homogenization? It can be a result of high oxygen pickup and/or inhomogeneity.
  • XRD lines should be marked with indexes on diffractograms.
  • XRD diffractograms are not XRD “patterns”. The scientific language should be used.
  • Since XRD peaks of alpha’, alpha” and beta phases are overlapped, there is no way to perform the quantitative phase analysis. Consequently Fig.2 is nice but has no sense.
  • Why do as-cast Ti-5Mo-10Nb alloy has HV about 500? It is too high for such Ti-based alloys and can be a result of high amount of oxygen in composition.
  • Discussion section is missing: after Results section, authors placed Conclusions.
  • Conclusion #1 is not enough sufficient due to impossibility to perform a quantitative phase analysis.
  • Conclusion #2 is too obvious. It is well-known that the mechanical properties of Ti-based alloys are structure sensitive and can be controlled by thermomechanical treatment
  • Conclusion #3 has no logic. The mechanical properties can be a result of presence of omega phase, but not vice versa.
  • Conclusions #4 and #5 are not correct. The Young modulus is important but not only the one property important for biomechanical compatibility.

Author Response

We would like to thank the reviewers and Academic Editor for their careful reading and suggestions. We seek to meet all of them, which will enrich the paper. The pertinent changes can be found in the revision mode in the text. Below we are answering all presented questions.

The manuscript is “some” research on the “effect of some thermomechanical treatments” of Ti-5Mo-Nb. The figures are well-designed, however all level of research and manuscript is very low. It is not clear why do authors use exactly these compositions, these experiments performed and why these conclusions are important for the readers. The authors are recommended to repeat the experiment with good quality ingots and rewrite the article and resubmit it next time. Before the resubmission, please clarify the following drawbacks:

Thank you very much for the revision. Indeed, it will improve the manuscript.

The scientific language should be used for the manuscript. For example the title now has word “some” instead of “various” and list of all activities performed by the authors, without any scientific significance and reason: “Effect of some thermomechanical treatments on microstructure, phase composition, Vickers microhardness, and Young's modulus of Ti-5Mo-Nb alloys for biomedical applications“.

We made a complete revision of the paper, improving the language. The changes to the title were accepted.

Since the alloys contain up to 30% of Nb, they should be named Ti-xNb-5Mo. It is common to mention the alloying elements with higher amount first.

Suggestion accepted.

It is not described in introduction why do authors use alloys with exactly from 0 to 30% of Nb. Impurities and compositions of ingots are not studied.

All the analysis of the as-cast alloys were presented in the paper "Preparation and characterization of novel as-cast Ti-Mo-Nb alloys for biomedical applications”, by Giovana Collombaro Cardoso, Gerson Santos de Almeida, Dante Oliver Guim Corrêa, Willian Fernando Zambuzzi, Marília Afonso Rabelo Buzalaf, Diego Rafael Nespeque Correa and, Carlos Roberto Grandini, submitted to Scientific Reports journal. This paper developed a new set of as-cast Ti-Mo-Nb alloys for applications as metallic biomaterials. The chemical composition, structure, microstructure, microhardness, and cytotoxicity were analyzed. The results showed the excellent quality of the ingots produced and that the constituent elements of the samples are close to the nominal compositions initially proposed. No agglomerates or segregated elements were observed, indicating good homogeneity of the ingots produced. The microstructure of the alloys showed to be sensitive to the addition of Nb: the amount of β phase increased as the Nb content of the alloys increased, having the Ti-5Mo-30Nb alloy presented only this phase. The Vickers microhardness values decreased with the increase of Nb, except for the Ti-5Mo-10Nb, which had its microhardness increased, probably due to the presence of the ω phase in its microstructure. Cytotoxicity tests show that the alloys have no cytotoxic effect and keep the cells viable, causing the stimulation for cell adhesion, which indicates that the alloys have great potential as biomaterials to be used in the health area.

Low-quality alloying elements are used (for example CP-Ti). It makes sense for large ingots, but not for 60g laboratory ingots. New pure ingots should be melted and studied. The impurities level and uniformity of composition should be evaluated.

It is not what the results of the chemical analysis of the ingots after melting showed. The impurities level is extremely low, as observed in the chemical analysis presented in the paper cited above. If necessary, we can send these results to the reviewer.

Why do XRD lines are so wide in Fig.1 especially after annealing and homogenization? It can be a result of high oxygen pickup and/or inhomogeneity. XRD lines should be marked with indexes on diffractograms. XRD diffractograms are not XRD “patterns”. The scientific language should be used. Since XRD peaks of alpha’, alpha” and beta phases are overlapped, there is no way to perform the quantitative phase analysis. Consequently Fig.2 is nice but has no sense.

The XRD diffractograms were explained more in the paper, with the phases marked in the figure. Due to this overlapping of the peaks associated with each phase, we used an analysis of the diffractogram using Rietveld’s Method, which has been largely used since 1970. The sheets of each phase were used from Inorganic Crystal Structure Database (ICSD).

Why do as-cast Ti-5Mo-10Nb alloy has HV about 500? It is too high for such Ti-based alloys and can be a result of high amount of oxygen in composition.

A possible explanation for this effect was introduced in the paper.

Discussion section is missing: after Results section, authors placed Conclusions.

The correct is the Results and Discussion Section. We apologize for this.

Conclusion #1 is not enough sufficient due to impossibility to perform a quantitative phase analysis.

Conclusion #2 is too obvious. It is well-known that the mechanical properties of Ti-based alloys are structure sensitive and can be controlled by thermomechanical treatment

Conclusion #3 has no logic. The mechanical properties can be a result of presence of omega phase, but not vice versa.

Conclusions #4 and #5 are not correct. The Young modulus is important but not only the one property important for biomechanical compatibility.

We re-written the conclusions.

Sincerely yours,

Prof. Carlos Roberto Grandini, FBSE

Round 2

Reviewer 1 Report

Dear Authors.

Thank you for answering the comments.

The manuscript can be accepted as it is.

Thank You.

Author Response

Thank you very much for your respect and acceptance of our paper.

Reviewer 3 Report

Unfortunately the authors formally answered some of my previous comments.  The level of the manuscript is still very weak and does not deserve the publication in a high impact journal such as Metals. The authors are welcome to read the material science articles on Ti-based alloys for biomedical application to increase their level of understanding how the material science article should be written. The following comments are not answered properly:

  • It is still not clear why do authors use alloys with exactly from 0 to 30% of Nb. Furthermore some binary and ternary Ti-Nb-based alloys exhibit superelasticity and extremely low Elastic modulus about 40 GPa and lower.
  • Impurities and compositions of ingots are not studied. If such results exist, they must be presented in the article.
  • There is still no explanation of extremely wide XRD lines in Fig.1 especially after annealing and homogenization. The lines width is not even measured.
  • The Rietveld refinement is not enough accurate for quantitative phase analysis of textured samples and for alpha”-phase with preferable orientation within parent beta phase especially if some lines of phases are overlapping. This makes impossible to perform quantitative phase analysis. Only quantitative analysis can be performed.
  • Conclusion are mostly well-known before this article, not supported by the discussion in the text or not enough sufficient:
  • Conclusion #1 is not still enough sufficient due to impossibility to perform a quantitative phase analysis.
  • Conclusion #2 is not complete. What does mean “more sensitive”? It is not a scientific language.
  • Conclusion #3 is not correct since the wear resistance has not been studied or even discussed in the article.
  • Conclusion #5 is not correct. There are commercial alloys with low or comparable modulus. Furthermore there is some developed alloys with almost twice lower modulus, The comparable alloys must be named.
  • Conclusion #6 has no sense. Always some alloys are better, some are worse. Since the s too obvious. It is well-known that the mechanical properties of Ti-based alloys are structure sensitive and can be controlled by thermomechanical treatment
  • Conclusions #7 is still not correct. Now it is specified to “studied mechanical properties”, however it makes the reader to search for what is the “studied mechanical properties” in the text.  And still the Young modulus is important but not only the one property important for biomechanical compatibility.

Author Response

Unfortunately the authors formally answered some of my previous comments. The level of the manuscript is still very weak and does not deserve the publication in a high impact journal such as Metals. The authors are welcome to read the material science articles on Ti-based alloys for biomedical application to increase their level of understanding how the material science article should be written.

We want to thank the reviewer for spending your precious time analyzing your paper.

The following comments are not answered properly:

It is still not clear why do authors use alloys with exactly from 0 to 30% of Nb. Furthermore some binary and ternary Ti-Nb-based alloys exhibit superelasticity and extremely low Elastic modulus about 40 GPa and lower.

As reported earlier and reinforced in the revised version of the paper, the Nb concentrations were chosen to study alloys with α+β phases, up to the alloy with only the β phase. It was not interesting to us included alloys exhibit superelasticity.

Impurities and compositions of ingots are not studied. If such results exist, they must be presented in the article.

As mentioned in the response in the earlier paper (in analysis for publishing in Scientific Reports journal), the chemical analysis was obtained and is reported below. We do not have authorization from the journal to reproduce the results in this paper before its publication.

Alloy

Ti (wt%)

Mo (wt%)

Nb (wt%)

Ti-5Mo

93.8 ± 0.4

6.2 ± 0.4

-

Ti-5Mo-10Nb

83.5 ± 0.9

5.3 ± 0.5

11.0 ± 0.6

Ti-5Mo-20Nb

73.3 ± 0.9

5.2 ± 0.9

21.3 ± 0.3

Ti-5Mo-30Nb

62.8 ± 0.5

5.8 ± 0.7

31.2 ± 0.4

It can be observed that the alloys presented compositions close to the nominal values with the presence of metallic impurities in nonsignificant concentrations. The sum of all possible impurities and oxygen and nitrogen are below 0.2 wt%. The melting technique with the inert argon-controlled atmosphere has been satisfactorily performed regarding the possible oxygen presence since titanium is highly reactive with oxygen.

It is important to emphasize that the ASTM F 2066–13 states that the composition of molybdenum to the Ti–15Mo alloy is between 14.00 and 16.00 wt %. In this way, the prepared alloy used in this article followed this standard and, therefore, was suitable for the study. Technical standards have not been established for the other Ti–15Mo–Nb alloys but the technical standards of other titanium alloys present tolerance of 1.0 Wt %. Thus, it can be concluded that all alloys were prepared according to the anticipated stoichiometry. Regarding the analysis of interstitial elements, which was based on the standard for Ti–15Mo,21, the tolerance values for oxygen and nitrogen were (0.20 6 0.02) and (0.05 6 0.02) Wt %, respectively. It can be observed concerning the oxygen content that all alloys were within the tolerance of this standard.

We also made EDS the elemental mappings of each produced alloy after melting. It is observed that the elements are well distributed. No agglomerated and segregated elements were observed, showing the excellent homogeneity of the ingots.

There is still no explanation of extremely wide XRD lines in Fig.1 especially after annealing and homogenization. The lines width is not even measured.

We do not have an explanation for the lines extremely wide. Probably the incorrect calibration of the equipment in this measurement.

The Rietveld refinement is not enough accurate for quantitative phase analysis of textured samples and for alpha"-phase with preferable orientation within parent beta phase especially if some lines of phases are overlapping. This makes impossible to perform quantitative phase analysis. Only quantitative analysis can be performed.

Due to this overlapping of the peaks associated with each phase, we used an analysis of the diffractogram using Rietveld’s Method, which has been largely used since 1970. The sheets of each phase were used from Inorganic Crystal Structure Database (ICSD).

To our knowledge, we do not have restrictions to use Rietveld’s analysis to obtain the analysis of the quantitative phase in Ti alloys (see for example, http://dx.doi.org/10.2451/2019PM870, http://dx.doi.org/10.1063/1.4707920, and https://www.malvernpanalytical.com/en/products/measurement-type/phase-quantification).

Conclusion are mostly well-known before this article, not supported by the discussion in the text or not enough sufficient:

Conclusion #1 is not still enough sufficient due to impossibility to perform a quantitative phase analysis.

As to our knowledge and supported by Rietveld’s method's possible use in phase quantification, we confirm this conclusion.

Conclusion #2 is not complete. What does mean "more sensitive"? It is not a scientific language.

We improved it.

Conclusion #3 is not correct since the wear resistance has not been studied or even discussed in the article.

Usually, alloys with an increase in the hardness improve the wear resistance (we observed this improvement in a paper in preparation).

Conclusion #5 is not correct. There are commercial alloys with low or comparable modulus. Furthermore there is some developed alloys with almost twice lower modulus, The comparable alloys must be named.

Of course, We agree. But new alloys with low cost can be developed. It is research.

Conclusion #6 has no sense. Always some alloys are better, some are worse. Since the s too obvious. It is well-known that the mechanical properties of Ti-based alloys are structure sensitive and can be controlled by thermomechanical treatment

Conclusions #7 is still not correct. Now it is specified to "studied mechanical properties", however it makes the reader to search for what is the "studied mechanical properties" in the text. And still the Young modulus is important but not only the one property important for biomechanical compatibility.

We improved them.